# MicroRNA-675-5p Overexpression Is an Independent Prognostic Molecular Biomarker of Short-Term Relapse and Poor Overall Survival in Colorectal Cancer

**DOI:** 10.3390/ijms24129990

**Published:** 2023-06-10

**Authors:** Spyridon Christodoulou, Christina D. Sotiropoulou, Panteleimon Vassiliu, Nikolaos Danias, Nikolaos Arkadopoulos, Diamantis C. Sideris

**Affiliations:** 1Fourth Department of Surgery, University General Hospital “Attikon”, National and Kapodistrian University of Athens, 12462 Athens, Greece; spyridon.christodoulou@yahoo.gr (S.C.); pant.greek@gmail.com (P.V.); ndanias@med.uoa.gr (N.D.); 2Department of Biochemistry and Molecular Biology, Faculty of Biology, National and Kapodistrian University of Athens, 15701 Athens, Greece; xristswt@biol.uoa.gr (C.D.S.); dsideris@biol.uoa.gr (D.C.S.)

**Keywords:** colon cancer, microRNA, molecular tumor markers, prognosis, prognostic biomarkers, small non-coding RNA

## Abstract

Colorectal cancer (CRC) is the main cause of cancer-related deaths globally, highlighting the importance of accurate biomarkers for early detection and accurate prognosis. MicroRNAs (miRNAs) have emerged as effective cancer biomarkers. The aim of this study was to investigate the prognostic potential of miR-675-5p as a molecular prognostic biomarker in CRC. For this reason, a quantitative PCR assay was developed and applied to determine miR-675-5p expression in cDNAs from 218 primary CRC and 90 paired normal colorectal tissue samples. To assess the significance of miR-675-5p expression and its association with patient outcome, extensive biostatistical analysis was performed. miR-675-5p expression was found to be significantly downregulated in CRC tissue samples compared to that in adjacent normal colorectal tissues. Moreover, high miR-675-5p expression was associated with shorter disease-free (DFS) and overall survival (OS) in CRC patients, while it maintained its unfavorable prognostic value independently of other established prognostic factors. Furthermore, TNM stage stratification demonstrated that higher miR-675-5p levels were associated with shorter DFS and OS intervals, particularly in patients with CRC of TNM stage II or III. In conclusion, our findings suggest that miR-675-5p overexpression constitutes a promising molecular biomarker of unfavorable prognosis in CRC, independent of other established prognostic factors, including TNM staging.

## 1. Introduction

The majority of colorectal cancer (CRC) cases can be attributed to preventable risk factors, including smoking, being overweight or eating poorly, being inactive, and drinking excessively. The epidemiology of CRC differs between various geographical locations, as well as between various racial groups, genders, and ages. However, it is the leading cause of mortality in men under the age of 50 and ranks second overall in cancer-related deaths [1,2]. Progressive genetic mutations and related tissue damage characterize the multiyear, multistage, and multipath processes of CRC carcinogenesis. Adenocarcinoma is the histological subtype of colorectal cancer that is most frequently found. The effective treatment of patients with colorectal adenocarcinoma depends on early diagnosis, as well as the early detection of recurrence [3,4].

The spectrum of therapy options for early stages and advanced diseases has expanded as a result of our growing understanding of the pathophysiology of CRC. In addition to the use of palliative chemotherapy, immunotherapy, and targeted therapy, the treatment options for CRC encompass a range of approaches. These include endoscopic and surgical local excision, the administration of preoperative radiation and systemic therapy to reduce the tumor stage, extensive surgery for cases of locoregional and metastatic disease, and local ablative therapies for treating metastases [5]. The highest chance of survival remains for people with non-metastasized cancer, even if these new treatment choices have increased the overall survival for advanced disease. Additionally, the sensitivity and specificity of the tumor markers that are now available for the detection of CRC are very low [6]. Their usage is restricted to tracking patient responses to treatment and detecting relapses following surgery [7,8]. Therefore, it is essential to find new, trustworthy biomarkers for the early diagnosis, precise staging, and follow-up of CRC progression [9].

Non-coding RNAs (ncRNAs) have gathered much attention recently for their role in the control of gene expression, making them intriguing molecular markers for a number of human diseases [10,11]. It is well established that ncRNAs can control the pathogenesis of CRC in a variety of ways [12]. Specifically, ncRNAs have been classified into distinct types based on their length, structure, and location. The four main ncRNA types with various roles in malignancies are microRNA (miRNA), long ncRNA (lncRNA), circular RNA (circRNA), and PIWI-interacting RNA (piRNA) [13,14,15,16]. By opening a window into the influence of the non-coding regions of the genome, the discovery of ncRNAs has brought a new dimension to our understanding of how cancer arises and how it may be treated [17,18]. miRNAs are by far the species of ncRNAs that have been the subject of the greatest research in cancer. This ncRNA type offers a potent new path for the identification of novel genetic risk factors for cancer and plays a crucial role in influencing molecular and cellular processes of the cancer state [19].

The non-coding RNA molecules known as miRNAs are short (19–23 nucleotides) and operate as internal regulators of gene expression by binding to the 3′-UTR of target mRNAs to cause translational repression or mRNA cleavage. On rare occasions, miRNAs may attach to the 5′-UTR of target mRNAs or even the coding sequence of mRNAs, acting as activators of gene expression [13]. The Hippo, Notch, and WNT/β-catenin signaling pathways are only a few of the many pathways that miRNAs control, and their participation in CRC is crucial [20,21]. *APC* or *CTNNB1* mutations typically result in the buildup of β-catenin, stimulating the WNT/β-catenin pathway and driving the evolution of CRC. For instance, it was reported that *APC* is a target of miR-135a, miR-135b, miR-942, and miR-494. *APC* is impacted adversely by miR-942, and miR-942 overexpression accelerates CRC development by inducing WNT/β-catenin signaling. Similar to this, miR-494 inhibits *APC*, causing-catenin to build up. Additionally, miRNAs control epithelial-to-mesenchymal transition (EMT) transcription factors, such as E-cadherin, N-cadherin, ZEB, SNAI1, and TWIST1, which can either promote or prevent the migration and invasion of different tumor cell types, including CRC [22].

Recent studies have highlighted the role of lncRNA H19 in the WNT/β-catenin pathway [23]. H19 can act as a miRNA sponge or epigenetic modulator, and it is a tank for miRNA-675 (miR-675-5p and miR-675-3p). By controlling numerous pathways involved in cell proliferation, EMT, and WNT/β-catenin signaling, miR-675-5p seems to have a crucial role in CRC. In particular, *APC*, *GSK3*, and *TCF4* are among the WNT/β-catenin pathway targets of miR-675-5p, according to reports [24,25,26]. Furthermore, higher levels of miR-675-5p have been associated with increased sensitivity to chemotherapy drugs, such as 5-fluorouracil (5-FU) [27,28].

Prompted by these findings, we aimed to investigate the expression levels and potential molecular biomarker utility of miR-675-5p in CRC. Thus, we attempted to quantify this miRNA in 218 primary CRC tissue samples and 90 paired, adjacent normal colorectal tissue samples, which were tested using an *in-house*-developed quantitative PCR (qPCR) assay. Then, we conducted thorough biostatistics analyses of the potential relevance of miR-675-5p as a prognostic molecular tumor biomarker.

## 2. Results

### 2.1. The Expression Levels of miR-675-5p Are Lower in CRC Tissue Specimens Than in Adjacent Non-Cancerous Colorectal Tissue Specimens

The median age of CRC patients included in the current study was 67.5-year-old (range: 35–93; interquartile range: 58–75). Table 1 lists the clinicοpathological features of the patients that were included in this study. Most of the CRC patients had a tumor located in the colon. Among the 218 CRC patients, over half of them had received chemotherapy, while most of the patients were classified in histological grade II and TNM stage III groups.

miR-675-5-p levels in CRC tissues samples ranged from 0.61 to 48,781 RQU with a mean ± SEM of 2542 ± 496.7, while in non-cancerous tissues, the levels ranged from 0.71 to 81,457 RQU with a mean ± SEM of 5721 ± 1728 (Table 2). When comparing the distribution in the two cohorts, it was evident that miR-675-5p levels were significantly downregulated (*p* < 0.001) in most of the malignant tumors, when compared to their paired non-cancerous samples (76 out of 90 paired tissues; 84.4%) (Figure 1).

### 2.2. High Levels of miR-675-5p Predict an Unfavorable Outcome for CRC Patient Survival

The next step included examining any correlation between miR-675-5p levels and the DFS and OS of CRC patients. Regarding the DFS analysis, 176 CRC patients that did not present with distant metastasis were included. Among these, 62 patients (64.8%) were diagnosed with tumor recurrence during the follow-up intervals. For the OS analysis, 203 CRC patients with complete available follow-up data were included, while 94 CRC-related deaths (53.7%) were recorded within this cohort.

When conducting survival analysis, we observed that CRC patients with high levels of miR-675-5p had significantly lower DFS probabilities (*p* < 0.001) compared to patients with lower miR-675-5p levels (Figure 2A). Moreover, the unfavorable prognostic value of miR-675-5p was observed in OS analysis as well, where CRC patients with higher miR-675-5p levels had significantly shorter OS (*p* < 0.001) intervals (Figure 2B). These results were also confirmed through univariate bootstrap Cox regression analysis, which showed an HR of 3.18 (*p* = 0.001) for disease recurrence in CRC patients with high miR-675-5p levels (Table 3) and an HR of 4.28 (*p* = 0.001) for death associated with CRC, in comparison to the patient group with lower levels of miR-675-5p (Table 4).

### 2.3. miR-675-5p Expression Retains Its Unfavorable Prognostic Significance Independently of Other Eshtablished Prognostic Factors

Multivariate Cox regression models were adjusted for the tumor location, histological grade, and TNM stage, as well as for the type of treatment received after tumor resection (radiotherapy or chemotherapy). Next, bootstrapping in Cox regression was performed with 1000 samples and BCa 95% CIs, ensuring accurate HRs in the regression analysis. From the multivariate Cox regression analysis, it was evident that the significance of the miR-675-5p expression status for the prognosis of the patients’ DFS remained unaffected (HR = 3.07; bootstrap *p* < 0.001) when combined with the status of the aforementioned clinicopathological factors (Table 3). Moreover, miR-675-5p retained the prognostic significance regarding OS as well (HR = 3.87; bootstrap *p* < 0.001) when combined with the tumor location, histological grade, TNM stage status, and whether the patients had received radiotherapy or chemotherapy (Table 4).

### 2.4. miR-675-5p Is an Unfavorable Prognostic Marker in Distinct Subgroups of CRC Patients

When stratifying CRC patients according to the tumor location, we noticed that patients with colon tumors (*n* = 125 out of 175 for DFS and *n* = 139 out of 203 for OS) and high levels of miR-675-5p showed significantly shorter DFS intervals (*p* = 0.020) and OS intervals (*p* < 0.001) (Figure 3A,B). Similarly, as shown in Figure 3C,D, CRC patients with rectum tumors (*n* = 50 out of 175 for DFS and *n* = 64 out of 203 for OS) and low levels of miR-675-5p had significantly higher probabilities for both DFS (*p* = 0.006) and OS (*p* = 0.002).

Furthermore, we stratified CRC patients according to the TNM stage. For the DFS and OS analysis, 85 CRC patients had TNM stage II disease and 66 had TNM stage III disease. According to the Kaplan–Meier curves, patients with TNM stage II and III and high levels of miR-675-5p had worse DFS (*p* = 0.005 and *p* = 0.030, respectively), in contrast to patients with the same TNM stages and low levels of miR-675-5p (Figure 4A,B). Regarding CRC patients with TNM stage II, their OS time intervals were also significantly shorter when they expressed miR-675-5p at high levels (*p* = 0.004) (Figure 4C). Similarly, we found that CRC patients with TNM stage III and high levels of miR-675-5p had worse OS, as well (*p* < 0.001) (Figure 4D). However, for patients with TNM stage I, there were no significant outcomes from this analysis.

## 3. Discussion

Colorectal cancer (CRC) is considered a major public health concern globally, mainly due to its high incidence and fatality rates [1]. It is characterized by the unregulated proliferation of malignant cells in the colon or rectum, which results in tumor formation [8]. CRC ranks among the most prevalent and lethal malignancies worldwide [1,2]. For this reason, the early detection of CRC is crucial for successful treatment and improved patient survival rates. The development of effective biomarkers for the early detection and accurate prognosis of CRC is critical for improving patient outcomes [7]. Several biomarkers are currently used in clinical practice, including carcinoembryonic antigen (CEA) and carbohydrate antigen 19-9 (CA 19-9), but their sensitivity and specificity remain inadequate [29,30]. As a result, there is an urgent need to discover novel and more effective biomarkers for CRC.

Non-coding RNAs, such as miRNAs, tRNA fragments, and long non-coding RNAs, including circular RNAs, comprise another large class of reliable molecular biomarkers in human malignancies [31], including CRC [32,33,34,35], yet none of all these putative cancer biomarkers have been prospectively validated in large studies regarding CRC patients and established in clinical routine, so far. Besides mRNAs that are known to constitute a rich source of potential biomarkers in CRC, with the prominent examples of kallikrein-related peptidases [36,37,38,39,40] and apoptosis-related [41,42,43] or stress-induced molecules [44,45], miRNAs have emerged as promising candidates for cancer biomarkers due to their involvement in various biological processes, including tumorigenesis, metastasis, and drug resistance [46,47]. These short non-coding RNA molecules influence gene expression at the post-transcriptional level and have been linked to the etiology of several malignancies, including colorectal cancer [22,48]. miRNAs exhibit unique expression profiles in different cancer types and stages, suggesting their potential as diagnostic and prognostic markers. In the context of CRC, identifying specific miRNAs that can accurately detect the disease at early stages and predict patient outcomes is of great interest [49,50,51,52,53,54,55]. A prevalent example is miR-675-5p, an miRNA with an established oncogenic role in CRC [56]. In particular, several studies have investigated the functional role of miR-675-5p in CRC, which is suggested to support hypoxia-induced EMT in colon cancer cells and promote CRC cell growth and drug resistance [24,26,27]. However, the clinical significance of miR-675-5p as a molecular marker has not been investigated yet. In this study, we examined the expression levels of miR-675-5p in CRC patient tumor samples and adjacent normal colorectal tissues, and we evaluated the prognostic potential of this miRNA as a biomarker for CRC.

Our research revealed that miR-675-5p levels in CRC tissues were much lower than those in corresponding non-cancerous tissues. The expression levels between the paired cancerous and non-cancerous specimens were significantly different. Increasing evidence and numerous studies have investigated the diagnostic value of miR-675-5p by evaluating its expression level. For example, in a study of miR-675-5p in paraffin-embedded breast cancer tissues, it was discovered that the expression of miR-675-5p was significantly higher in patients than in the control group, and there was no correlation between miR-675-5p expression levels and the age of patients or lymph node metastasis [57]. Additionally, miR-675-5p expression is significantly elevated in gastric cancer patients and could be a potential diagnostic marker [58]. It is important to note that in our study, according to the results of multivariate Cox regression analysis, the relevance of miR-675-5p expression status to patients’ DFS and OS prognosis remained unaffected when combined with the tumor location, histological grade, TNM stage status, and whether the patients had received radiotherapy or chemotherapy.

Based on our findings, poor prognosis for CRC patients is associated with high levels of miR-675-5p. The disease-free and overall survival intervals were significantly longer in patients with lower miR-675-5p levels. These data demonstrate that miR-675-5p could separate patients with a high risk of CRC recurrence and mortality. All of the above led to figuring out the potential of mir-675-5p as a prognostic biomarker in CRC. Furthermore, we investigated the prognostic value of miR-675-5p in CRC patients according to the TNM stage. We discovered that patients with TNM stage II had substantially shorter DFS and OS intervals when the miR-675-5p levels were elevated. In the same manner, the DFS and OS of patients with high levels of this miRNA and TNM stage III was worse, in comparison to patients to those in with low levels of miR-675-5p. Patients with CRC might have a wide range of prognoses, even within the same TNM stage. The DFS and OS of patients with TNM stage II-III CRC is affected by several factors besides tumor stage, such as tumor location, age, histological type, and sex [59]. As a result, miR-675-5p may be an inherent prognostic factor in different stages of CRC and could contribute to a better stratification system in CRC patients as an independent marker.

The distal colon, proximal colon, and rectum have historically been the three areas of the gut where CRCs have been described in the literature [60]. It should be emphasized that although the rectum belongs to the hindgut embryonically, it is occasionally characterized independently [61]. A variety of miRNAs have been characterized as tissue-specific biomarkers that improve the understanding of the molecular differences between these local tumors. For instance, hsa-miR-20a appears to have much higher expression levels in the rectum than in the colon. On the other hand, hsa-miR-145 has substantially higher expression levels in the colon than in the rectum [62]. In this study, higher expression levels of miR-675-5p in both the colon and rectum correlated with poor DFS and OS. Although this miRNA cannot be characterized as tissue-specific, the above evidence supports the idea that it is a promising prognostic biomarker independent of the tumor site. In CRC, it is also important to note that elevated serum miR-675-5p levels are linked to worse clinical outcomes [28], so the utility of miR-675-5p as a non-invasive biomarker is strengthened.

MiR-675-5p has an important role in the development and progression of several malignancies. By targeting tumor suppressors, the lncRNA H19-derived miR-675-5p is linked to the advancement of non-small cell lung cancer (NSCLC) [63]. In addition, miR-675-5p is associated with glycolysis reprogramming in oral cancer and is characterized as a hypoxia-regulated miRNA [64,65]. In this way, we were triggered to study its role in CRC survival outcomes. Costa et al. demonstrated that miR-675-5p is overexpressed in metastatic colon cancer cells, promoting EMT by controlling hypoxia-inducible factor 1 subunit alpha (HIF1a) [26]. In addition, the hypoxic tumor microenvironment in CRC induces the expression levels of miR-675-5p, thereby enhancing drug resistance [27]. It may seem counter-intuitive that miR-675-5p levels are downregulated in the majority of colorectal adenocarcinoma tissues compared to those in adjacent normal colorectal tissues, whereas miR-675-5p overexpression is being suggested as a predictor of adverse prognosis in this malignancy. Nonetheless, considering that miR-675-5p overexpression in CRC cells was shown to have an oncogenic role [56], it is tempting to speculate that an increase in miR-675-5p expression constitutes a late event in colorectal carcinogenesis. In fact, this miRNA could act as a double-edged sword in cancer cells. A similar contradiction constitutes miR-28-5p in CRC; in fact, although miR-28-5p is downregulated in colorectal tumors compared to their adjacent non-cancerous tissues, its high expression has been shown to predict the poor DFS and OS of colorectal adenocarcinoma patients, independently of clinicopathological prognosticators and standard patient treatment, including radiotherapy and chemotherapy [50]. Our findings highlight the significant role of miR-675-5p in the diagnosis and prognosis of CRC. The heterogeneity of this disease and the complicated molecular pathways associated with CRC progression render the discovery of novel biomarkers imperative. The unfavorable prognostic value of high miR-675-5p levels is important in clinical practice and may improve patient stratification.

Despite these encouraging results, it is important to note some limitations of this study. In order to highlight the molecular mechanism of miR-675-5p in CRC development and progression, in vivo experiments are required. Furthermore, validation based on a larger cohort with various clinical characteristics may provide a more comprehensive assessment of its biomarker utility. Last but not least, the heterogeneity of CRC urges the investigation of a panel of biomarkers, as a single biomarker might not adequately capture the complexities of CRC biology, which would restrict its therapeutic value.

## 4. Materials and Methods

### 4.1. Collection of Colorectal Tissue Samples

The colorectal tissue biobank used in this study included 218 primary colorectal adenocarcinoma specimens from patients who had surgery at the University General Hospital “Attikon” between 2000 and 2019. After the tumor was surgically removed, all cancer tissue samples were histologically evaluated by a pathologist and immediately frozen in liquid nitrogen. In 90 instances, a sample of normal colorectal tissue was also provided. This study was approved by the institutional ethics committee of the University General Hospital “Atikon” (approval number: 31; date 29 January 2009) and was conducted in accordance with the Helsinki Declaration. Each patient was informed about the study’s aims and consented to provide a sample.

### 4.2. Characteristics of Tumors and Survival Outcomes of the CRC Patient Cohort

The clinicopathological features documented and used in the current investigation included tumor size and location, histological grade, and TNM stage. The TNM classification includes tumor invasion (T), regional lymph node status (N), and the presence or absence of distant metastases (M). In addition to the events and cause of death, follow-up information included disease status (disease-free or recurrence), overall survival status (living or deceased), and disease status. Fifteen (15) of the 218 patients in our study were not included in the survival analysis because follow-up data were unavailable. Twenty-seven (27) patients out of the remaining 203 with complete follow-up data had distant metastasis (M1) at the time of tumor surgical excision and were thus excluded from the DFS analysis.

### 4.3. Cell Culture, Total RNA Isolation, In Vitro Polyadenylation, and cDNA Synthesis

The RPMI-1640 Medium was used to cultivate the human colorectal adenocarcinoma cell line DLD-1, which was modified to incorporate 10% fetal bovine serum, 100 kU/L penicillin, and 0.1 g/L streptomycin. According to the American Type Culture Collection (ATCC) recommendations, DLD-1 cells were cultured in an incubator at 37 °C with an adjusted CO_2_ concentration of 5%, before being trypsinized and collected for subsequent applications.

Next, a monophase solution of guanidine thiocyanate and phenol, which is distributed as TRI Reagent^®^ (Molecular Research Center, Inc., Cincinnati, OH, USA), was used to homogenize and dissolve all tissue samples. According to the manufacturer’s recommendations, DLD-1 cells and homogenized colorectal tissues were used to isolate total RNA, which was then diluted in THE RNA Storage Solution (Life Technologies Ltd., Carlsbad, CA, USA) and stored in a deep freezer at −80 °C. A BioSpec-nano Micro-volume UV-Vis Spectrophotometer was used (Shimadju, Kyoto, Japan) for the spectrophotometrical measurement of the concentration and purity of the RNA, at 260 and 280 nm.

The next steps included total RNA polyadenylation and reverse transcription using an oligo-d(T) adapter primer (sequence: 5′-GCGAGCACAGAATTAATACGACTCACTATAGGTTTTTTTTTTTTVN-3′) to generate first-strand cDNA [66]. More specifically, a recombinant E. coli poly(A) polymerase (New England Biolabs Ltd., Whitby, ON, Canada) was used to polyadenylate the miRNAs, and as a primer for first-strand cDNA synthesis, an oligo-dT adapter was utilized. The oligo-dT adapter primer provides a universal priming site, allowing for the successful amplification of small non-coding RNAs.

### 4.4. Primer Design and Real-Time Quantitative Polymerase Chain Reaction (qPCR)

In order to detect miR-675-5p, specific forward primers were designed for the reference genes SNORD43 (RNU43) and SNORD48 (RNU48) (Table 5). Each forward primer was combined with a universal reverse primer that matches the oligo-dT-adapter used during cDNA synthesis to amplify the target sequences in quantitative PCR (qPCR). The qPCR reaction mixture contained 0.5 μL of 10-fold diluted cDNA from both the DLD-1 cell line and patient samples, 5 μL of KAPA™ SYBR^®^ FAST qPCR master mix (2×) (Kapa Biosystems Inc., Woburn, MA, USA), 1 μL of each primer (final concentration: 200 nM), and 2.5 μL of RNase-free H_2_O. The qPCR experiment was performed on an ABI 7500 Fast Real-Time PCR System (Applied Biosystems™, Thermo Fisher Scientific, Waltham, MA, USA), following the standard thermal cycling and melting procedure described previously [66].

The comparative threshold cycle (2^−ΔΔCt^) method was used to calculate the ratio of miR-675-5p molecules to the geometric mean of SNORD43 and SNORD48 molecules, to normalize the qPCR assays for the amount of RNA added to reverse transcription reactions, divided by the same ratio for the DLD-1 cDNA, which was used as a calibrator to allow results from different RT-qPCR runs to be compared. The normalized data were presented in relative quantification units (RQUs). As previously mentioned, all conditions needed for the 2^−ΔΔCt^ method implementation were verified [67].

### 4.5. Biostatistics

The data analysis was conducted using IBM SPSS Statistics software (version 28) (IBM Corp., Armonk, NY, USA). The distribution of miR-675-5p levels in both non-cancerous and cancerous colorectal tissue samples deviated from a normal distribution. For this reason, non-parametric tests, such as the Mann–Whitney U test and Wilcoxon signed-rank test for paired samples, were employed to assess the significance of differences in miR-675-5p levels between these groups. Similarly, the Mann–Whitney U test or Kruskal–Wallis test was utilized to analyze variations in miR-675-5p levels among different patient subgroups based on clinicopathological features.

Next, Kaplan–Meier survival analysis was employed to investigate the association between patient outcomes, in terms of disease-free survival (DFS) and overall survival (OS), and miR-675-5p levels. Using the X-tile software (version 3.6.1), an appropriate prognostic cut-off value corresponding to the 31st percentile (4.00 RQU) was determined to categorize patients as high or low expressors of miR-675-5p. The Mantel–Cox (log-rank) test was utilized to assess differences in DFS and OS between the resulting Kaplan–Meier curves.

To validate the prognostic value of miR-675-5p levels and estimate the hazard ratio (HR) for disease recurrence and disease-related death, bootstrapped Cox regression analyses were performed with 1000 bootstrap samples for internal validation. The bootstrap bias-corrected and accelerated (BCa) method was used to calculate bootstrap *p*-values and 95% confidence intervals (CIs) for each estimated HR. Additionally, multivariate Cox regression models were constructed, adjusting for the most significant clinicopathological characteristics. The statistical significance level was set at *p*  <  0.050.

## 5. Conclusions

In conclusion, we demonstrate a statistically significant difference in the expression levels of miR-675-5p in CRC patients compared to those in non-cancerous adjacent tissues. We also demonstrated that miR-675-5p is an independent unfavorable prognostic biomarker for DFS and OS in CRC. Our results highlight the biological significance of miR-675-5p in CRC and raise the possibility of its utilization in clinical practice. Additional studies are necessary to thoroughly evaluate the potential of miR-675-5p as a prognostic biomarker in larger cohorts.

## Figures and Tables

**Figure 1 ijms-24-09990-f001:**
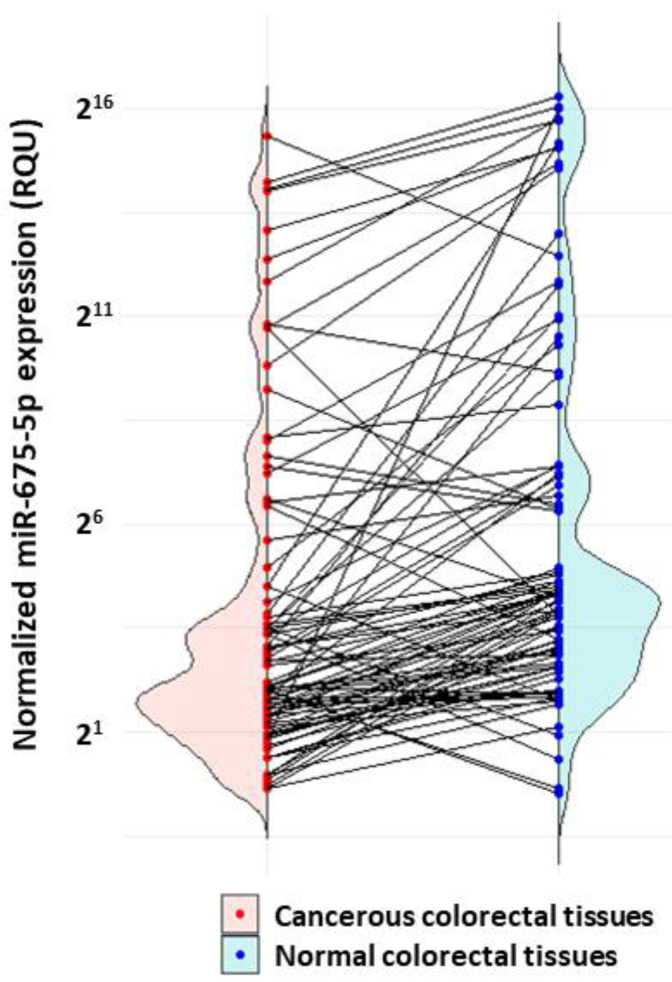
Split violins illustrating the miR-675-5p expression levels in cancerous vs. normal adjacent colorectal tissues, after comparing 90 pairs of tissue specimens. The miR-675-5p expression levels were lower in most colorectal tumors. The *p*-value was calculated using the Wilcoxon signed-rank test.

**Figure 2 ijms-24-09990-f002:**
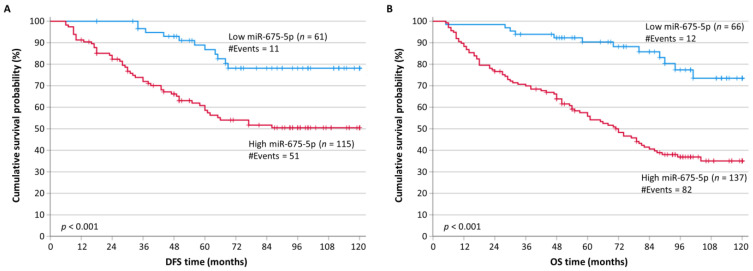
Kaplan–Meier survival curves for the disease-free survival (DFS) and overall survival (OS) of colorectal cancer (CRC) patients. Patients with tumors highly expressing miR-675-5p had significantly shorter DFS (**A**) and OS (**B**) time intervals than patients bearing tumors with low miR-675-5p levels. The *p*-values were calculated using the Mantel-Cox (log-rank) test.

**Figure 3 ijms-24-09990-f003:**
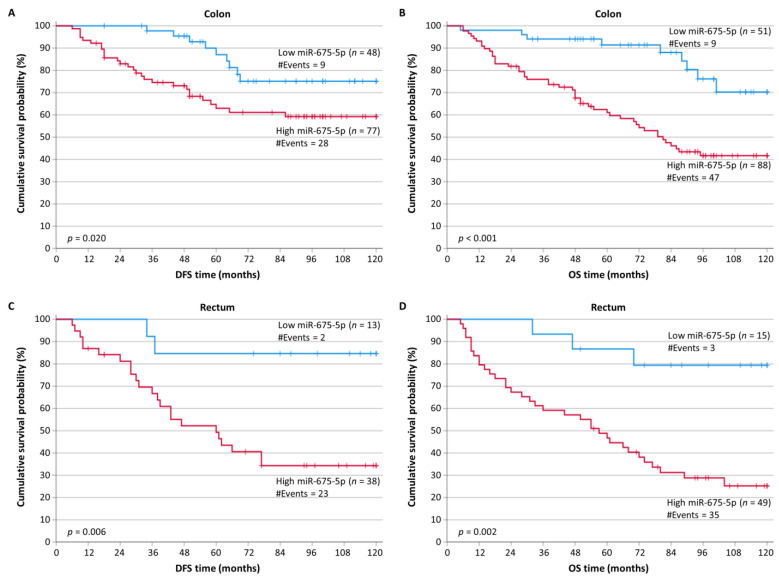
Stratified Kaplan–Meier survival curves for the disease-free survival (DFS) and overall survival (OS) of colorectal cancer (CRC) patients, according to tumor location. Patients with colon tumors that highly expressed miR-675-5p had shorter DFS (**A**) and OS (**B**) time intervals than patients bearing colon tumors with low miR-675-5p levels. Moreover, CRC patients with a rectum tumor location overexpressing miR-675-5p had poorer DFS (**C**) and OS (**D**) time intervals than patients with lower levels of miR-675-5p. The *p*-values were calculated using the Mantel-Cox (log-rank) test.

**Figure 4 ijms-24-09990-f004:**
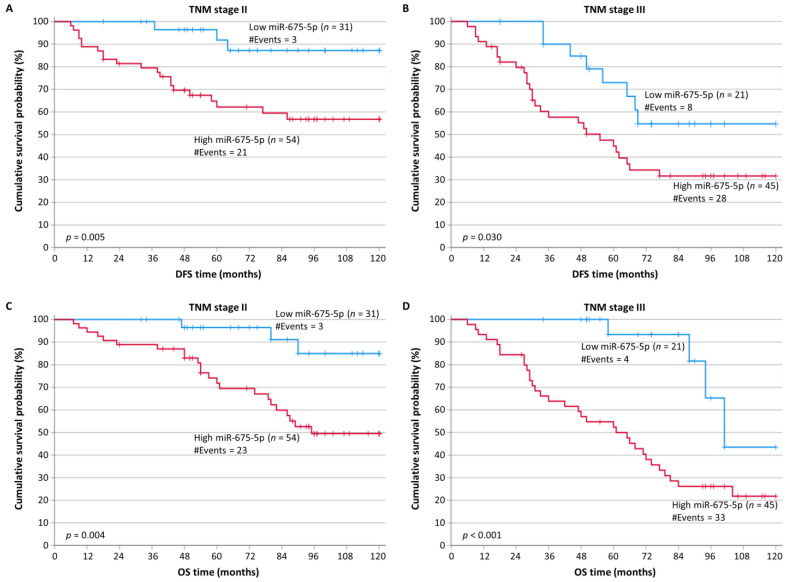
Stratified Kaplan–Meier survival curves for the disease-free survival (DFS) and overall survival (OS) of colorectal cancer (CRC) patients, according to TNM stage. Patients with tumors expressing a high level of miR-675-5p at TNM stages II (**A**) and III (**B**) had shorter DFS times than patients with tumors expressing a low level of miR-675-5p. Additionally, OS was worse for CRC patients with TNM stage II (**C**) or stage III (**D**) colorectal cancers overexpressing miR-675-5p compared to those with lower expression of this miRNA at the same TNM stage. The *p*-values were calculated using the Mantel–Cox (log-rank) test.

**Table 1 ijms-24-09990-t001:** Clinicopathological characteristics of the tumors of colorectal cancer (CRC) patients included in the current study.

	Number of Patients (%)
**Gender**	
Male	114 (52.3%)
Female	104 (47.7%)
**Tumor location**	
Colon	148 (67.9%)
Rectum	70 (32.1%)
**Histological grade**	
I	19 (8.7%)
II	167 (76.6%)
III	32 (14.7%)
**T (tumor invasion)**	
T1	6 (2.8%)
T2	26 (11.9%)
T3	135 (61.9%)
T4	51 (23.4%)
**N (nodal status)**	
N0	122 (56.0%)
N1	57 (26.1%)
N2	39 (17.9%)
**M (distant metastasis)**	
M0	191 (87.6%)
M1	27 (12.4%)
**TNM stage**	
I	28 (12.8%)
II	89 (40.8%)
III	74 (34.0%)
IV	27 (12.4%)
**Radiotherapy**	
No	169 (83.3%)
Yes	34 (16.7%)
**Chemotherapy**	
No	86 (42.4%)
Yes	117 (57.6%)

Abbreviation: TNM—tumor, node, and metastasis.

**Table 2 ijms-24-09990-t002:** miR-675-5p expression levels in tumors and normal colorectal tissue samples of colorectal cancer (CRC) patients.

Variable	Mean ± SEM	Range	Quartiles		
			1st	2nd (Median)	3rd
Normalized miR-675-5p expression (RQU)					
in cancerous tissues (*n* = 218)	2542 ± 496.7	0.61–48,781	3.27	20.74	692.4
in normal tissues (*n* = 90)	5721 ± 1728	0.71–81,457	7.03	19.65	170.7

Abbreviations: RQU, relative quantification units; SEM, standard error of the mean.

**Table 3 ijms-24-09990-t003:** miR-675-5p expression and disease-free survival (DFS) of colorectal cancer (CRC) patients.

	Covariate	HR ^1^	95% CI ^2^	*p*-Value ^3^	95% Bootstrap BCa ^4^ CI ^2^	Bootstrap *p*-Value ^3^
**Univariate analysis**	miR-675-5p expression					
	Low	1.00				
	High	3.18	1.66–6.11	<0.001	1.86–6.86	<0.001
	Tumor location					
	Colon	1.00				
	Rectum	1.75	1.06–2.91	0.030	1.02–2.98	0.031
	Histological grade					
	I	1.00				
	II	1.73	0.62–4.79	0.30	0.78–7.72	0.25
	III	2.83	0.90–8.90	0.075	1.01–18.43	0.049
	TNM stage					
	I	1.00				
	II	4.28	1.01–18.11	0.049	1.29–26.260	0.048
	III	9.61	2.31–40.00	0.002	2.97–67.669	0.004
	Radiotherapy					
	No	1.00				
	Yes	1.81	1.02–3.19	0.042	0.95–3.26	0.041
	Chemotherapy					
	No	1.00				
	Yes	2.33	1.33–4.07	0.003	1.39–4.42	0.004
**Multivariate (*n* = 176)**	miR-675-5p expression					
	Low	1.00				
	High	3.07	1.59–5.92	<0.001	1.66–8.02	<0.001
	Tumor location					
	Colon	1.00				
	Rectum	2.69	1.34–5.40	0.005	1.35–7.40	0.011
	Histological grade					
	I	1.00				
	II	1.06	0.37–3.08	0.91	0.36–5.77	0.92
	III	1.60	0.48–5.32	0.44	0.47–11.40	0.44
	TNM stage					
	I	1.00				
	II	4.89	1.10–21.72	0.037	1.46–39.676	0.025
	III	9.70	2.08–45.22	0.004	3.04–105.6	0.004
	Radiotherapy					
	No	1.00				
	Yes	0.49	0.22–1.09	0.081	0.16–1.12	0.10
	Chemotherapy					
	No	1.00				
	Yes	1.36	0.71–2.61	0.35	0.56–2.99	0.41

^1^ Hazard ratio, estimated from proportional hazard Cox regression. ^2^ Confidence interval of the estimated HR. ^3^ Statistically significant *p*-values are shown in italics. ^4^ Bias-corrected and accelerated.

**Table 4 ijms-24-09990-t004:** miR-675-5p expression and overall survival (OS) of colorectal cancer (CRC) patients.

	Covariate	HR ^1^	95% CI ^2^	*p*-Value ^3^	95% Bootstrap BCa ^4^ CI ^2^	Bootstrap *p*-Value ^3^
**Univariate analysis**	miR-675-5p expression					
	Low	1.00				
	High	4.28	2.33–7.85	<0.001	2.53–9.26	<0.001
	Tumor location					
	Colon	1.00				
	Rectum	1.62	1.07–2.45	0.022	1.03–2.45	0.020
	Histological grade					
	I	1.00				
	II	1.35	0.62–2.94	0.45	0.66–3.87	0.46
	III	2.53	1.06–6.02	0.036	1.06–9.05	0.047
	TNM stage					
	I					
	II	2.55	0.89–7.32	0.082	0.95–13.88	0.074
	III	5.73	2.03–16.16	<0.001	2.31–30.43	<0.001
	IV	35.24	11.88–104.5	<0.001	13.23–200.4	<0.001
	Radiotherapy					
	No	1.00				
	Yes	1.62	1.00–2.64	0.051	0.94–2.63	0.058
	Chemotherapy					
	No	1.00				
	Yes	1.84	1.19–2.85	0.006	1.21–2.95	0.007
**Multivariate (*n* = 203)**	miR-675-5p expression					
	Low	1.00				
	High	3.87	2.10–7.13	<0.001	2.42–7.95	<0.001
	Tumor location					
	Colon	1.00				
	Rectum	1.43	0.85–2.41	0.18	0.83–2.60	0.20
	Histological grade					
	I	1.00				
	II	0.76	0.33–1.75	0.52	0.26–2.49	0.58
	III	1.16	0.46–2.90	0.76	0.33–3.90	0.82
	TNM stage					
	I	1.00				
	II	2.88	0.97–8.53	0.056	1.05–19.54	0.045
	III	6.79	2.20–20.99	<0.001	2.33–73.65	<0.001
	IV	39.17	12.06–127.2	<0.001	14.27–417.8	<0.001
	Radiotherapy					
	No	1.00				
	Yes	1.08	0.57–2.02	0.82	0.53–2.01	0.83
	Chemotherapy					
	No	1.00				
	Yes	0.78	0.46–1.33	0.36	0.41–1.40	0.39

^1^ Hazard ratio, estimated from proportional hazard Cox regression. ^2^ Confidence interval of the estimated HR. ^3^ Statistically significant *p*-values are shown in italics. ^4^ Bias-corrected and accelerated.

**Table 5 ijms-24-09990-t005:** Primer pairs used in real-time quantitative PCR (qPCR).

Target	Primer Sequence (5′ to 3′)	Direction	Length (nt)	T_m_ (°C)
miR-675-5p	GAGGCCCGGGTTCGATTC	Forward	18	62
*SNORD43*	ACTTATTGACGGGCGGACA		19	59
*SNORD48*	TGATGATGACCCCAGGTAACTCT		23	59
None (Universal primer)	GCGAGCACAGAATTAATACGAC	Reverse	22	56

Abbreviations: nt, nucleotides; T_m_, melting temperature.

## Data Availability

The data presented in this study are available on reasonable request from the corresponding authors.

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
