# Peer review of "MicroRNA-675-5p Overexpression Is an Independent Prognostic Molecular Biomarker of Short-Term Relapse and Poor Overall Survival in Colorectal Cancer"

_ijms, 2023, doi:10.3390/ijms24129990_

Round 1

Reviewer 1 Report

A study by Christodoulou investigates the significance of microRNA-675-5p as an independent prognostic molecular biomarker of short-term relapse and poor overall survival in colorectal cancer. Overall, the manuscript must be presented clearly considering low input of experimental results. The presentation of the results should be particularly improved.

Specific comments:

1. In the Fig. 1, the Authors show one of the most important results in the study, however, the presentation should be improved. For clarity, the Authors should consider to e.g., change the type of graph. 

2. In Materials and Methods the Authors describe culturing of cell line DLD-1, however, I cannot find any results. 

3. Table 5 - why sequences for only forward primers have been given?

acceptable

Author Response

Reviewer #1 (Comments to the Author):

  1. In the Fig. 1, the Authors show one of the most important results in the study, however, the presentation should be improved. For clarity, the Authors should consider to e.g., change the type of graph.

We deeply appreciate the Reviewer’s suggestion to visualize our data in a different way. Therefore, we replaced our previous Figure 1 with a new one (split violin plot). This plot shows both distributions of expression values (paired cancer vs. normal samples). We sincerely hope that now the result is more clear for the reader.

  1. In Materials and Methods the Authors describe culturing of cell line DLD-1, however, I cannot find any results.

We thank the Reviewer for this comment. cDNA produced by reverse transcription of the polyadenylated DLD-1 total RNA extract was used as a calibrator sample for the needs of the comparative Ct (ΔΔCt) method of calculation of relative miR-675-5p expression; in the same context, SNORD43 and SNORD48 were both used as reference genes, again for expression normalization.

The determination of miR-675-5p expression in DLD-1 cells (normalized only against both reference genes) would not be correct, as equal amounts of all three amplicons do not generate equal fluorescence signals. In fact, the SYBR Green dye (used in the real-time qPCR) is not used in a saturating dye concentration for maximal fluorescence signal; therefore, different amplicons, with distinct lengths and % GC amount cannot be directly compared to achieve relative quantification.

In conclusion, there are no results concerning the DLD-1 cell line, as its cDNA was used for technical reasons of qPCR-based calculations based on the ΔΔCt method.

  1. Table 5 - why sequences for only forward primers have been given?

With all due respect to the Reviewer’s comment, this is not correct; Table 5 includes the common (universal) reverse qPCR primer used for amplification of the target (miR-675-5p) and both reference genes (SNORD43 and SNORD48), besides the forward specific primers. However, prompted by the Reviewer’s comment, we found that we had not previously included in the text the sequence of the oligo-dT-adaptor primer, used during first-strand cDNA synthesis, after polyadenylation of the total RNA extracts; its sequence now appears embedded in the text (Page 12). Part of this adaptor is the annealing site for the universal reverse primer, used for real-time qPCR.

The authors wish to thank the Reviewers for their constructive comments that led to the improvement of the current manuscript.

Reviewer 2 Report

This study reports the analysis of mir-675-5p as a potential biomarker for CRC. The study is quite limited since it is purely observational, however the biomarker analysis is thorough with good prospective follow-up of patients so I think this could warrant publication. I have a couple of comments/queries:

- If mir-675-5p plays an oncogenic role in CRC (as alluded to in discussion line 232, lines 282-291), it makes sense for it to be associated with poorer outcome (figure 2) - but why is it higher in normal tissue vs cancer tissue (table 2)? This seems counter-intuitive, and questions the role and thus potential of this miRNA as a cancer-associated marker.

- A couple of paired specimens in figure 1 seem higher in cancer tissue vs normal tissue, contrary to mean levels across the cohort in table 2. Did the authors explore any variables linked to these patients? When the authors say that they looked at high miR-675-5p in tumors (figure 2 legend), is this tumor rather than normal tissue, in which the miR-675-5p is higher?

- There is no age/sex information for the patients, which I would normally expect to see, did the authors collect this information and can they confirm that no differences were observed between these groups?

Author Response

Reviewer #2 (Comments to the Author):

  1. If mir-675-5p plays an oncogenic role in CRC (as alluded to in discussion line 232, lines 282-291), it makes sense for it to be associated with poorer outcome (figure 2) - but why is it higher in normal tissue vs cancer tissue (table 2)? This seems counter-intuitive, and questions the role and thus potential of this miRNA as a cancer-associated marker.

We would like to thank the Reviewer for this comment. We added the following in the manuscript, to clarify this seemingly paradoxical result:

Page 11 (Discussion): It may seem counter-intuitive that miR-675-5p levels are downregulated in the majority of colorectal adenocarcinoma tissues compared to adjacent normal colorectal tissues, whereas miR-675-5p overexpression is being suggested as a predictor of adverse prognosis in this malignancy. Nonetheless, considering that miR-675-5p overexpression in CRC cells was shown to exert an oncogenic role [56], it is tempting to speculate that an increase in miR-675-5p expression constitutes a late event in colorectal carcinogenesis. In fact, this miRNA could act as a double-edged sword in cancer cells. A similar contradiction constitutes miR-28-5p in CRC; in fact, although miR-28-5p is downregulated in colorectal tumors compared to their adjacent non-cancerous tissues, its high expression has been shown to predict poor DFS and OS of colorectal adenocarcinoma patients, independently of clinicopathological prognosticators and standard patient treatment, including radiotherapy and chemotherapy [50]. Our findings highlight the significant role of miR-675-5p in the diagnosis and prognosis of CRC. The heterogeneity of this disease and the complicated molecular pathways associated with CRC progression render the discovery of novel biomarkers imperative. The unfavorable prognostic value of high miR-675-5p levels is important in clinical practice and may improve patients’ stratification.

  1. A couple of paired specimens in figure 1 seem higher in cancer tissue vs normal tissue, contrary to mean levels across the cohort in table 2. Did the authors explore any variables linked to these patients? When the authors say that they looked at high miR-675-5p in tumors (figure 2 legend), is this tumor rather than normal tissue, in which the miR-675-5p is higher?

Prompted by the Reviewer’s suggestion, we explored all available variables for these patients overexpressing miR-675-5p in their tumors compared to matched (adjacent) normal colorectal mucosa; no pattern or association was found between miR-675-5p and any tested variable.

Regarding the categorization into high and low miR-675-5p–expressing samples, this was based on the optimal cutoff value selected for prognostic purposes using the X-tile software, as stated in the Materials and Methods (Page 13). This normalized expression value of miR-675-5p was equal to 4.00 RQU 31st percentile of the distribution of miR-675-5p expression values in tumor samples). After this dichotomization of miR-675-5p expression, we proceeded with Kaplan-Meier and Cox regression analyses. Thus, no normal tissue samples were taken into consideration for Figure 2.

Finally, it should be noted that all samples (both the normal and cancerous tissue part of the samples) were histologically confirmed by a pathologist after tumor resection, as clearly stated in the Materials and Methods (Page 11).

  1. There is no age/sex information for the patients, which I would normally expect to see, did the authors collect this information and can they confirm that no differences were observed between these groups?

Following the Reviewer’s suggestion, we added descriptives regarding the gender of CRC patients in Table 1. Moreover, we added the information regarding patients’ age (mean, range, interquartile range) in the text:

Page 3 (Results): The median age of CRC patients included in the current study was 67.5 years old (range: 35 – 93; interquartile range: 58 – 75).

No statistically significant differences regarding the gender or age of CRC patients were found between those with high and those with low expression of intracellular miR-675-5p expression.

The authors wish to thank the Reviewers for their constructive comments that led to the improvement of the current manuscript.

Round 2

Reviewer 1 Report

All comments have been adequately addressed.